# AXL Knock-Out in SNU475 Hepatocellular Carcinoma Cells Provides Evidence for Lethal Effect Associated with G2 Arrest and Polyploidization

**DOI:** 10.3390/ijms222413247

**Published:** 2021-12-09

**Authors:** Tugce Batur, Ayse Argundogan, Umur Keles, Zeynep Mutlu, Hani Alotaibi, Serif Senturk, Mehmet Ozturk

**Affiliations:** 1Izmir Biomedicine and Genome Center, Dokuz Eylul University Health Campus, Izmir 35330, Turkey; tugce.batur@ibg.edu.tr (T.B.); ayse.argundogan@msfr.ibg.edu.tr (A.A.); umur.keles@ibg.edu.tr (U.K.); zynpmtl86@gmail.com (Z.M.); hani.alotaibi@ibg.edu.tr (H.A.); 2Izmir International Biomedicine and Genome Institute, Dokuz Eylul University, Izmir 35330, Turkey; 3School of Medicine, Dokuz Eylul University, Izmir 35340, Turkey; 4Galen Research Center, Izmir Tinaztepe University, Izmir 35400, Turkey

**Keywords:** AXL, hepatocellular carcinoma, CRISPR-Cas9, gene knock-out, cell cycle arrest, DNA damage, polyploidy

## Abstract

AXL, a member of the TAM family, is a promising therapeutic target due to its elevated expression in advanced hepatocellular carcinoma (HCC), particularly in association with acquired drug resistance. Previously, RNA interference was used to study its role in cancer, and several phenotypic changes, including attenuated cell proliferation and decreased migration and invasion, have been reported. The mechanism of action of AXL in HCC is elusive. We first studied the AXL expression in HCC cell lines by real-time PCR and western blot and showed its stringent association with a mesenchymal phenotype. We then explored the role of AXL in mesenchymal SNU475 cells by CRISPR-Cas9 mediated gene knock-out. AXL-depleted HCC cells displayed drastic phenotypic changes, including increased DNA damage response, prolongation of doubling time, G2 arrest, and polyploidization in vitro and loss of tumorigenicity in vivo. Pharmacological inhibition of AXL by R428 recapitulated G2 arrest and polyploidy phenotype. These observations strongly suggest that acute loss of AXL in some mesenchymal HCC cells is lethal and points out that its inhibition may represent a druggable vulnerability in AXL-high HCC patients.

## 1. Introduction

Hepatocellular carcinoma (HCC) affects more than 900,000 people each year, and it is the third most common cause of cancer-related deaths worldwide [1]. HCC patients have a low survival rate (median of survival is 9 months) and this tumor is extremely resistant to systemic chemotherapy [2]. Among solid cancers, HCC has one of the fewest druggable genetic alterations, limiting treatment options for advanced HCC [3]. Although, seven drugs for targeted therapy of HCC have been approved, their efficacy is limited. For example, systemic treatments with sorafenib, lenvatinib, regorafenib, cabozantinib, or ramucirumab provide median overall survival (mOS) rate ranging between 8.5 to 13.5 months [4].

A newly approved drug for second-line therapy, Cabozantinib, inhibits the activity of several receptor tyrosine kinases, including VEGFR, MET, and AXL, a member of the TAM (TYRO3, AXL, MER) family [5]. TAM family of receptors are expressed selectively in different normal cells, in particular, macrophages, and mediate the clearance of dead cells by efferocytosis [6,7]. For reasons not wholly understood, AXL is overexpressed in multiple solid cancers, including HCC, particularly during EMT (epithelial-mesenchymal transition), and acquired drug resistance. AXL is a critical player in cancer cell EMT [8,9] and the development of tumor metastasis and drug resistance in HCC [8,10,11]. It has recently been firmly established that AXL is the master regulator of EMT in HCC cells [8].

Previous reports that addressed the effects of AXL inactivation in different cancer cell models were based on knock-down by RNA interference or kinase activity inhibition by different small chemicals [10,11,12,13,14,15]. The main concern with these approaches is the inability to completely deplete AXL expression or fully inhibit its activity. Therefore, the effects observed under such conditions may reflect the consequences of a down-regulation of AXL activity rather than the lack of expression. This aspect is particularly important for mesenchymal HCC cell lines that overexpress AXL more than 100-fold compared to epithelial-like HCC cell lines associated with more than 15-fold upregulation of active phospho-AXL levels [8]. In order to achieve complete suppression of AXL expression, the best available method is gene inactivation by CRISPR/Cas9 targeting. However, this method appears to be highly ineffective for AXL inactivation in cancer cells. Indeed, to our knowledge, there is only one published example of studies related to phenotypic effects of AXL knock-out in the literature which has been reported in a mouse breast cancer cell line [16]. A previous attempt to inactivate the AXL gene by CRISPR/Cas9 technique in the SNU449 HCC cell line was unsuccessful [17]. Here we describe the successful inactivation of AXL expression in SNU475, a mesenchymal-like HCC cell line. We extensively characterized these AXL knock-out cells and have obtained new critical data. We demonstrate that the AXL expression is indispensable for SNU475 cell line to avoid DNA damage and cell cycle arrest leading to lethal polyploidization and loss of tumorigenicity.

## 2. Results

### 2.1. AXL Has Elevated Expression in Mesenchymal-like HCC Cell Lines

Based on previous studies associating the induction of AXL expression during EMT in cancer cells and reported studies on some HCC cell lines [10,11,18,19], we decided to compare the relative expression of AXL in a large panel of HCC cell lines. As shown in Figure 1, we grouped our HCC cell lines as epithelial-like and mesenchymal-like according to previously reported classifications [8,20]. AXL transcripts and protein levels were analyzed by real-time PCR (qPCR) and Western Blot techniques, respectively (Figure 1). It was previously described that the AXL gene encodes two alternative spliced isoforms differing from each other by the presence or absence of exon 10 [21]. We called full-length form AXL FL and the exon 10 missing short variant form AXL V (Figure 1A). In order to explore which of the isoforms is predominant in HCC cells, qPCR targeting exon 9–10 for AXL FL and exon 10–11 junction for AXL V were carried out. qPCR data are presented in Figure 1B,C for FL and V forms, respectively. Although the expression of the longer transcript (AXL FL) and its shorter counterpart (AXL V) was correlated, AXL FL was expressed at a higher rate than AXL V in mesenchymal-like HCC cell lines, Mahlavu, Focus, SNU449, SNU475, and SNU387. Both isoforms were undetected in epithelial-like HCC lines SNU398, Huh7, Hep3B, and HepG2. Of note, weakly positive AXL expression was detected in Hep40 cell line (Figure 1B,C, compare the y axis of two panels). At the protein level, increased expression was detected in mesenchymal-like HCC cell lines (Mahlavu, Focus, SNU182, SNU423, SNU449, SNU475, and SNU387). In contrast, epithelial-like HCC cell lines (HepG2, Hep3B, Hep3B-TR, Huh7, Hep40, and SNU398) displayed no or low expression, consistent with the results of qPCR for FL AXL mRNA (Figure 1D).

### 2.2. Successful Knockout of AXL Gene in SNU475 HCC Cell Line

To elucidate the function of AXL in HCC, we performed CRISPR-Cas9-mediated AXL knock-out in Focus, Mahlavu, SNU449, and SNU475 cells lines that overexpress AXL, as demonstrated in Figure 1. To this end, we used two single guide RNAs (herein dubbed gRNA), AXL-45 and AXL-185, targeting AXL gene locus at two different sites (exon 1 and exon 2; shown in Appendix A) and non-targeting Renilla gRNA (specific to Renilla luciferase gene) served as the negative control. Similar to the previously reported findings in SNU449 cell line [17], AXL gene knock-out also proved relatively inefficient in all four HCC cell lines tested in this study. As compared to Renilla gRNA, AXL-185 gRNA caused a moderate decrease in AXL protein levels. In contrast, experiments with AXL-45 gRNA were more efficient, causing strong inhibition of expression as tested by western blot, albeit a total loss was not reached (Appendix A). We noticed that the AXL expression in AXL gRNA-targeted cells was progressively returned initial levels in successive passages during cell culture (For example, Appendix A for SNU449 cells, other cell lines not shown). This observation strongly suggested that cells that lost AXL expression suffer a survival disadvantage in the cell culture at the expense of AXL-expressing cell population. Therefore, we decided to increase our chances of obtaining AXL knockout cells by single cell cloning. Since AXL depletion was best achieved in SNU475 cells with AXL-45 gRNA (Appendix A), we continued our experiments with this setup. Since the backbone plasmid of gRNAs had Venus gene (Appendix A), Venus positive cells were sorted as single-cell using a cell sorter (Appendix A–D). Single cell-derived clones were grown in culture and tested for AXL expression (Figure 2A). Many clones expressed low but detectable AXL expression. The expression of AXL was near background levels in clones c17 and c18 (Figure 2A). We could not expand the c18 clone, so we had to continue our further analyses with clone 17, hereafter called AXL-c17 (Figure 2A).

### 2.3. Augmented Nucleus Size and Different Cellular Morphology of AXL Knock-Out SNU449 Clone AXL-c17

In cell culture, AXL-c17 showed an aberrant change in cellular morphology compared with the control REN-4 clone. Therefore, we stained the nuclei of cells with DAPI (4′,6-diamidino-2-phenylindole) and calculated nuclear area by Image J program (Figure 2B). The average nuclear area of REN-4 control cells was about 200 μm^2^. In contrast, we measured the average nuclear area of AXL knock-out AXL-c17 cells close to 600 μm^2^ (Figure 2C). Thus, approximately a three-fold increase of nuclear area was detected in AXL-c17 cells (*p* < 0.0001) (Figure 2B,C). To further illustrate the differences in cellular morphology between AXL-c17 and REN-4 control cells, we studied the cytoskeletal arrangement by phalloidin staining and nuclear integrity by lamin B staining. As a result, AXL-c17 was frequently associated with increased nuclear size, accompanied by multinuclearity (Figure 2D).

### 2.4. AXL Knock-Out Cells Exhibit Cell Cycle Aberrations, Polyploidy, and Low Clonogenicity

To further explore the effects of AXL depletion in SNU475 cells, cell cycle analyses were carried out at 72 h post-seeding. Based on DNA content analysis, cell cycle distribution in control REN-4 cells was as expected, with the majority of cells being detected at the G1 phase (>60%), followed by S and G2 phases. Sub-G1 and polyploid (>G2) cells were both less than 3%. In sharp contrast, however, AXL knockout c17 cells displayed gross cell cycle aberrations. Specifically, G1 cells dropped to less than 10%, while G2 phase cells rose to almost 40%. More interestingly, more than one-third of these AXL-depleted cells displayed polyploidy, as manifested by pseudotetraploidy. Finally, there was no significant difference in the number of sub-G1 cells (Figure 3A,B).

In accordance with the abundant polyploidy, AXL-c17 cells displayed crippled cell proliferation, as manifested by the formation of mini colonies (Figure 3C,D; *p* < 0.0001). Based on these observations, we calculated the duplication times of REN-4 and AXL-c17 cells in culture. Control cells had an average duplication time of 45.6 h, whereas AXL-deficient c17 cells performed the same task at a mean rate of 75.9 h.

### 2.5. G2 Arrest and Polyploidization of AXL Knock-Out Cells Is Associated with Features of Unresolved DNA Damage Response

Upon exposure to DNA damaging agents, cells expressing mutant p53 fail to undergo apoptosis or cease proliferation by arresting at G1 phase, and instead, they arrest at the G2 checkpoint [22,23,24]. Moreover, such G2-arrested cells can enter endoreduplication cycles resulting in the formation of polyploid cells over time [25]. Based on the fact that parental SNU475 cells display multiple p53 mutations [26], we hypothesized that the G2 arrest and polyploidization that we observed in these cells following genetic ablation of AXL gene was a manifestation of DNA damage response. To test our hypothesis, we performed a series of analyses to explore DNA damage response in AXL knock-out cells compared to control cells. DNA damage response is a complex process, which involves damage sensors (such as ATM and ATR) and mediators (such as 53BP1 and BRCA1), transducers such as Chk1 and Chk2, and effectors such as p53 and Cdc25 [27]. We first examined the number of nuclear foci decorated with phospho-ATM (Ser 1981) and 53BP1 by immunofluorescence and counted them (Figure 4A,B). Most of the control REN-4 cells displayed less than 10 foci per nucleus, whereas AXL knock-out c17 cells generally displayed more than 10 foci per nucleus (Figure 4A). Similar results were observed with 53BP1-positive foci. Specifically, AXL-c17 cells displayed mostly more than six foci, unlike REN-4 cells, which displayed less than 6 foci (Figure 4B).

Another outcome of DNA damage response is cellular senescence that can also occur after G2 arrest [22,28]. To elucidate the confounding accumulation in the G2 phase of the cell cycle, we tested whether cells entered senescence. HCC derived epithelial-like cell line, Huh7, was used as a positive control, and senescence was induced by treating cells with 50 and 100 nM doxorubicin. Senescence morphology was observed in positive control Huh7 cells but not in doxorubicin-treated and non-treated SNU475 REN-4 and AXL-c17 cells, indicating that G2 arrest was not caused by cells entering senescence (Appendix A).

Findings reported here with AXL-c17 clone may represent the cellular effects of AXL gene inactivation and off-target effects of the CRISPR-Cas9 system. Therefore, we sought to obtain additional AXL knock-out clones from SNU475 cell line. We performed additional viral infection experiments on SNU475 cells using lentiviral vectors expressing gRNA, namely AXL-45 and AXL-185. Despite being a challenging task, after long efforts, we managed to derive only two additional clonal cell lines, namely 3-12C from AXL-45 and 6-10H from AXL-185 infections (Figure 5A,B). First, the lack of AXL expression was demonstrated by flow cytometry (Figure 5A) and western blot analyses (Figure 5B). Of note, all bands in REN-4 were related to AXL since we did not observe those in AXL knock-out clones Figure 5B). Previously generated AXL-c17 clone served as a control in these assays. Then, cell cycle analysis in early passage cells was shown in Figure 5C. These studies consistently yielded decreased G1 phase cells associated with a reciprocal increase in G2 phase and polyploid cells, but an increase in S phase cells was also detected. Compared to G1 phase cells observed at 67% in REN-4 cells, AXL-3-12C and AXL-6-10H cells displayed 33% and 30% G1 phase cells, respectively. This was accompanied by increased S phase (26–27% from 17%), G2 phase (15–20% from 8%), and polyploid cells (12–21% from 2%), as shown in Figure 5C.

Much like other previously generated AXL knock-out clones, the AXL-3-12C isogenic clone failed to survive during extended periods of cell culture. Thus, we were able to perform additional experiments with AXL-6-10H and AXL-c17 clones only in comparison with REN-4 cells. Clonogenicity of these isogenic cell lines was assessed by plating flow cytometry sorted single cells into 96-well plates and following their survival and proliferation for 15 days using Cell Metric equipment. Representative examples of colony growth patterns are shown in Figure 6. REN-4-derived single-cell colonies displayed logarithmic growth, forming robust clonal groups populated with more than 200 cells (Figure 6A). In contrast, single cells derived from both AXL knock-out clones failed to form such colonies most of the time (Figure 6B,C), with rare exceptions. We tried to expand three different FACS-sorted single cell-derived clones from each subline, but we were successful with two AXL-c17 and only one AXL-6-10H-derived clones. Cell cycle analysis was performed on these clones to test whether they recovered from the aberration observed before single cell-sorted cloning. As shown in Figure 6D, AXL knock-out cells remained growth-defective with G2 arrest and polyploidy.

### 2.6. Induction of G2 Arrest and Polyploidization by Pharmacological Inhibition of AXL in SNU475 Cells

To confirm that the G2 arrest and polyploidization observed in AXL knock-out cells were indeed due to AXL loss of function, we treated the parental SNU475 cell line with a selective small molecule inhibitor of AXL, R428 (BGB324, Bemcentinib) [29]. The IC50 of parental cells was calculated as 3.2 μM (Figure 7A). Parental SNU475 cells were treated for 24, 48, and 72 h with 1.5, 3, and 6 µM R428 and subjected to cell cycle analysis. As shown in Appendix A and Figure 7B, we observed a dose-dependent increase in the percentage of G2 phase and polyploid cells upon treatment with R418. This confirmed that these phenotypic changes were indeed due to pharmacological inhibition of AXL.

### 2.7. AXL Knock-Out Cells Lost Their Tumorigenic Ability

Following in vitro experiments that showed the critical role of AXL in maintaining proliferation ability of mesenchymal-like SNU475 cells, we performed in vivo tumorigenicity studies as shown in Figure 8, using a total of ten immunodeficient NOD-SCIDγ mice, as described in Materials and Methods sections. Out of ten animals, one was deceased early in the experiment and therefore excluded from the study. As shown in Table 1, and Appendix A, control REN-4 cells were tumorigenic in all five animals tested. In sharp contrast, we detected no tumor formation in any of the four animals injected with AXL knock-out AXL-c17 cells after 10 weeks of follow-up (*p* = 0.0042). As REN-4 clones were fully tumorigenic and AXL-c17 clones were not tumorigenic at all, we stopped in vivo studies at this stage for ethical reasons.

## 3. Discussion

AXL is one of the several receptor kinases inhibited by Cabozantinib, a drug recently approved for the treatment of advanced HCCs [5]. The role of AXL inhibition in the efficacy of Cabozantinib treatment is not known. However, AXL is overexpressed in a subset of HCCs [11,12,18,30,31]. Its expression in clinical samples correlated with advanced tumor stage, multiple tumors, a higher incidence of vascular invasion, recurrence of disease, and mortality [10,18,32]. In addition, serum soluble AXL protein levels were reported as a new biomarker for HCC [33,34,35]. Phenotypic effects of AXL expression in HCC cell lines have been studied by RNA interference-mediated knock-down experiments. Downregulation of AXL expression attenuated cell proliferation, migration, and invasion in vitro and inhibited tumor metastasis in vivo [11,18,19,36]. Thus, the dramatic effects of AXL knock-out in SNU475 cells that we report here have not been observed before. RNA interference is based on the downregulation of mRNA, leading to a decrease in transcript levels instead of complete loss of function [37].

Our results indicate that AXL expression is closely correlated with a mesenchymal phenotype in HCC cell lines (Figure 1). This close correlation conforms with a recently published report demonstrating that the expression of AXL is the main driver of EMT in HCC cells [8]. The same study also reported that AXL is the most induced and most active tyrosine kinase expressed in mesenchymal HCC cells. Crippled survival of AXL knock-out SNU449 cells that is observed here, as opposed to previously reported mild growth inhibitory effects, is probably a result of our experimental approach aiming at inactivation of AXL rather than downregulating its expression. This severe response appears to be due to increased DNA damage response in these cells in the absence of any external induction of DNA damage (Figure 4). Thus, in the absence of AXL, SNU475 cells are not able to deal with appropriate repair of DNA damage either because of increased damage load and/or decreased repair capacity. Increased DNA damage response following AXL inhibition has been reported for breast, ovarian, and lung cancer cell lines [14,38,39]. Here, we provide evidence that AXL knock-out alone is sufficient to block the survival of cancer cells by a mechanism associating G2 arrest with polyploidization (Figure 3, Figure 5 and Figure 6). G2 arrest and polyploidization have been observed in p53-deficient cancer cells following exposure to DNA damaging agents [25], although this has also been reported for p53-positive cells [40]. As SNU475 cell line displays multiple mutations, including p53 mutations [20,26], it will be interesting to further explore the mechanisms of polyploidization in these cells. However, this will be extremely difficult since these cells grow very poorly in cell culture (Figure 6). Finally, we show that R428, a well-known inhibitor of AXL kinase activity, also induced G2 arrest and polyploidy in SNU475 cells, albeit less severely (Figure 7). Since this small chemical is quite active on different HCC cell lines, its growth inhibition could be mediated by G2 arrest and polyploidization as described here for SNU475 cells.

We would like to conclude by stressing that our observations about AXL dependency of mesenchymal HCC cells are based on a single cell line. As displayed in Figure 2A, our attempts to obtain additional AXL-knockout clones from several other mesenchymal HCC cell lines were unsuccessful. Therefore, it is presently unknown whether AXL dependency for survival is restricted to SNU475 cells or applicable to additional mesenchymal HCC cell lines. Further investigations are required for a better understanding of AXL roles in mesenchymal HCC cells.

## 4. Materials and Methods

### 4.1. Cell Culture

Epithelial-like (Hep3B, Hep3B-TR, Huh7, HepG2, Hep40, SNU398) and mesenchymal-like (Mahlavu, Focus, SNU387, SNU182, SNU423, SNU449, SNU475) HCC cell lines were used in the experiments [8,41,42]. The authenticity of these cell lines was confirmed by STR analysis. All cell lines were maintained in complete RPMI medium (11875168, Thermo Fisher Scientific, Waltham, MA, USA) supplemented with 10% FBS (16000036,Thermo Fisher Scientific, Waltham, MA, USA), 1% penicillin-streptomycin (15140148,Thermo Fisher Scientific, Waltham, MA, USA ), and 1% non-essential amino acid mixture (10270106, Thermo Fisher Scientific, Waltham, MA, USA) at 37 °C with 5% CO_2_.

### 4.2. Antibodies

Anti-AXL (C89E7, Cell Signaling, Danvers, MA, USA), anti-β-actin (8226, Abcam, Cambridge, MA, USA) antibodies were used in western blot assays. Lamin B (M-20,Santa Cruz Biotechnology, Dallas, TX, USA), Alexa Fluor 568 Phalloidin (A12380, Thermo Fisher, Waltham, MA, USA), 53BP1 (4937S, Cell Signaling, Danvers, MA, USA), and p-ATM Ser-1981 (D25E5, Cell Signaling, Danvers, MA, USA) were used in immunofluorescence, and PE-conjugated anti-human AXL antibody (FAB154P, R&D Systems, Minneapolis, MN, USA) was used in the flow cytometry.

### 4.3. Cloning

pECPV (EFS-Cas9-P2A-Venus) vector was used as the backbone plasmid in CRISPR-Cas9 knock-out experiments. This backbone was derived from lenticrispr V2 vector (Addgene plasmid #52961) by successfully replacing Puromycin resistance coding sequence with the Venus gene, which encodes an improved version of the yellow fluorescent protein (YFP). Details on the creation and design of the backbone were described previously [43]. Cloning procedure was carried out as described by Shalem et al. [44]. Briefly, pECPV vector was digested with BsmBI enzyme (ER0451, Thermo Fisher, Waltham, MA, USA) and dephosphorylated with FastAP (EF0654, Thermo Fisher, Waltham, MA, USA). Oligonucleotides used in the cloning of gRNAs (listed in Appendix A) were phosphorylated by T4 PNK (M0201S, NEB, Ipswich, MA, USA) and annealed using a thermal cycler program set to 30 min at 37 °C, 5 min at 95 °C, and ramp down to 25 °C at 5 °C/min. Annealed and phosphorylated oligonucleotides and digested backbone plasmid were ligated with Quick Ligase (M2200S, NEB, Ipswich, MA, USA) and transformed into Stbl-3 bacteria.

### 4.4. Viral Packaging and Transduction

Packaging of the viral particles was accomplished by co-transfecting with packaging plasmids pLP1, pLP2, and pVSVG (Invitrogen, Carlsbad, CA, USA) and gRNA containing pECPV vectors into Hek-293T packaging cells using Lipofectamine 2000 reagent (Thermo Fisher, Waltham, MA, USA). The supernatants containing the viral particles were collected and filtered at 48 h post-transfection. Cells were transduced with the viral particles at a 1/20 dilution. Three days after transduction, Venus signal was assessed using BD LRS Fortessa flow cytometry (BD Bioscience, San Jose, CA, USA).

### 4.5. Obtaining Monoclones

Venus positive cells were sorted into 96-well plates as single-cell/well, using BD Cell Sorter ARIAIII (BD Bioscience, San Jose, CA, USA). The growth of cell colonies was routinely controlled under a microscope, and when sufficient growth was observed, they were transferred into 6-well plates and so on.

### 4.6. Western Blot

When the confluency reached about 70–80%, the cells were detached with trypsin and rinsed with PBS, and the cell pellets were collected by centrifugation. After discarding the supernatant, the pellets were lysed in a modified RIPA buffer (150 mM NaCl, 1% Nonidet P-40, 0.5% Sodium deoxycholate, %0.1 SDS, and 25 mM pH7.4 Tris, 1 mM PMSF, 1 mM NaF, 1 mM Na_3_VO_4_). For protein extraction, the tubes were kept on ice for 20–30 min by occasional vortexing every 3–4 min. The tubes were then centrifuged at 15,000× *g* for 30 min, and protein extracts were quantified using the Pierce BCA Assay Kit (ThermoFisher, Waltham, MA, USA). Total protein samples were denatured in 4× loading dye containing 20% of β-mercaptoethanol at 95 °C. Protein samples and protein size markers were then loaded on SDS-PAGE gel composed of 8% or 10% resolving and 5% stacking gels. Initially, 70 V was applied to the gel cast to allow the samples to pass through the stacking gel. Then, the proteins were run on the resolving gel with a voltage of 110 V for 2 h. Proteins were wet-transferred onto PVDF membranes (Immobilon-P, Millipore, Billerica, MA, USA) for 1.5 h at 400 mA. Membranes were blocked by using 5% non-fat milk. After blocking, the membranes were blotted with 1/1000 diluted primary antibody overnight at 4 °C and then 1/1000 diluted secondary antibody for 1 h at RT.

### 4.7. Immunofluorescence

Cells were seeded onto appropriate coverslips assembled in 24-well plates. The next day, cells were fixed with 3.7% formaldehyde diluted in PBS. Fixed cells were blocked for 1 h in PBS containing 0.1% Triton X-100 and 10% FBS. All primary antibodies, diluted to 1/200 in blocking solution, were added, and the coverslips were incubated at 4 °C overnight. The next day, the primary antibody was rinsed thoroughly, and the cells were incubated with a species-matching fluorescent secondary antibodies at a ratio of 1/1000 dilution for 1 h at RT. After rinsing three times with PBS, DAPI staining was performed, and coverslips were mounted onto slides using a mounting medium.

### 4.8. Colony Formation

1500 cells per well were seeded in a 12-well cell culture dish, and colony growth was followed for 3 weeks. Cell culture growth medium was renewed once every 3 days. At the end of 3 weeks, the growth medium was completely removed, and then the cells were washed once with 1X PBS followed by fixation with ice-cold 100% methanol for 5 min at −20 °C. After discarding the fixative, colonies were stained with 0.5% (*w*/*v*) crystal violet dye at room temperature for 20 min.

### 4.9. Cell Cycle

Cells were incubated at 37 °C with complete RMPI in all cell cycle experiments for 72 h unless stated otherwise. Parental SNU-475 cells were incubated with RMPI alone or with 1.5, 3, and 6 μM R428 at 37 °C for 24, 48, and 72 h. Cell cycle analysis was performed as described briefly herein. First, cells were trypsinized and collected by centrifugation. Cell pellets were then rinsed and resuspended in PBS and fixed with ice-cold ethanol, added dropwise while vortexing the samples to achieve 70% final concentration. Cells were initially kept on ice for 15 min and then at −20 °C overnight. The next day, fixed cells were rinsed with PBS and resuspended in PI staining solution containing 50 µg/mL PI, 1 mg/mL RNAase A, 0.05% Triton X. Staining was performed at 37 °C for 45 min. PI staining was evaluated by BD LRS Fortessa (BD Bioscience, San Jose, CA, USA). Analysis was performed by FlowJo v10 (Flowjo, OH, USA) with Watson Pragmatic algorithm.

### 4.10. Flow Cytometry

Cells were collected by trypsinization and washed with PBS. Then, they were incubated with 1/10 PE-conjugated AF154 (FAB154P) antibody in PBS containing 10% FBS for 30 min at room temperature. After washing three times with PBS, cells were dissolved in FACS buffer (10% FBS in PBS), and the PE signal was measured by BD LRS Fortessa.

### 4.11. Quantitative Real-Time PCR (qPCR)

Total RNA isolation was carried out with the Nucleospin RNA II kit (Macherey-Nagel, Düren, Germany), and 1 μg of total RNA was used for cDNA synthesis using RevertAid First Strand cDNA Synthesis Kit (Thermo Scientific, Waltham, MA, USA). Quantitative real-time PCR was performed using 1/10 diluted cDNA, the TaqMan Universal master mix II and Universal Probe Library probes (Roche, Basel, Switzerland) in the ABI PRISM 7500 Fast qPCR System (Thermo Scientific, Waltham, MA, USA) with reaction conditions; 95 °C for 15 min, 50 cycles of 95 °C for 15 s and 60 °C for 1 min. Primers and probes are shown in Appendix A. Fold changes of gene expressions were calculated by ΔΔCt method according to GAPDH, which was used as a control for mRNA quantity.

### 4.12. Clonality-Proliferation Assay

Venus positive cells were sorted in a 96-well plate as a single cell/well by BD Cell Sorter ARIAIII. Growth of cell colonies was imaged by Solentim Cell Metric (Solentim, Dorset, UK)

### 4.13. Sulforhodamine B Assay

3 × 10^3^ cells per well were seeded into 96-well plates. The next day, cells were treated with serially diluted (final concentration: from 10μM to 0.312μM) AXL inhibitor R428 (HY-15150, MedChemExpress, Princeton, NJ, USA) dissolved in DMSO. After 72 h of incubation, cells were fixed with 10% ice-cold trichloroacetic acid (TCA) (T6399, Sigma-Aldrich, St. Louis, MO, USA) and incubated at −20 °C for 20 min. Fixed cells were stained with 0.4% Sulforhodamine B (SRB) (S1402, Sigma-Aldrich, St. Louis, MO, USA) in 1% acetic acid (CAS:64-19-7, Sigma-Aldrich, St. Louis, MO, USA ) solution for 20 min. After staining, excess SRB was removed with 1% acetic acid, and the plates were left to dry. The absorbance in each well was measured after adding 50 μL of 10 mM Tris-Base (T1503, Sigma-Aldrich, St. Louis, MO, USA). The OD was measured at 512 nm using Varioskan, FLASH (Thermo Fisher Scientific, Waltham, MA, USA).

### 4.14. Animal Studies

All animal experiments were approved by the IBG Animal Experiments Local Ethics Committee (IBG-AELEC) with protocol number 21/2016. Six to seven months-old NOD-SCIDγ male mice bred in the same cage were randomly chosen for injection of REN-4 and AXL-c17 cells, respectively. Initially, respective clones were injected at left and right back (1 × 10^6^ cells for each injection) subcutaneously, using one mouse for each clone. For the following experiments, we performed a single injection of 2 × 10^5^ cells per animal. A total of eight animals (four animals for each clone) were used. Overall, in vivo tumorigenic studies used ten animals. Before injection, cells were harvested in a sub-confluent state (70–80%) and resuspended in 50 μL PBS and then mixed with 50 μL growth factor-reduced matrigel (356230, Corning, NY, USA). Blinded tumor measurements were carried out twice a week by a technician of IBG vivarium facility.

## Figures and Tables

**Figure 1 ijms-22-13247-f001:**
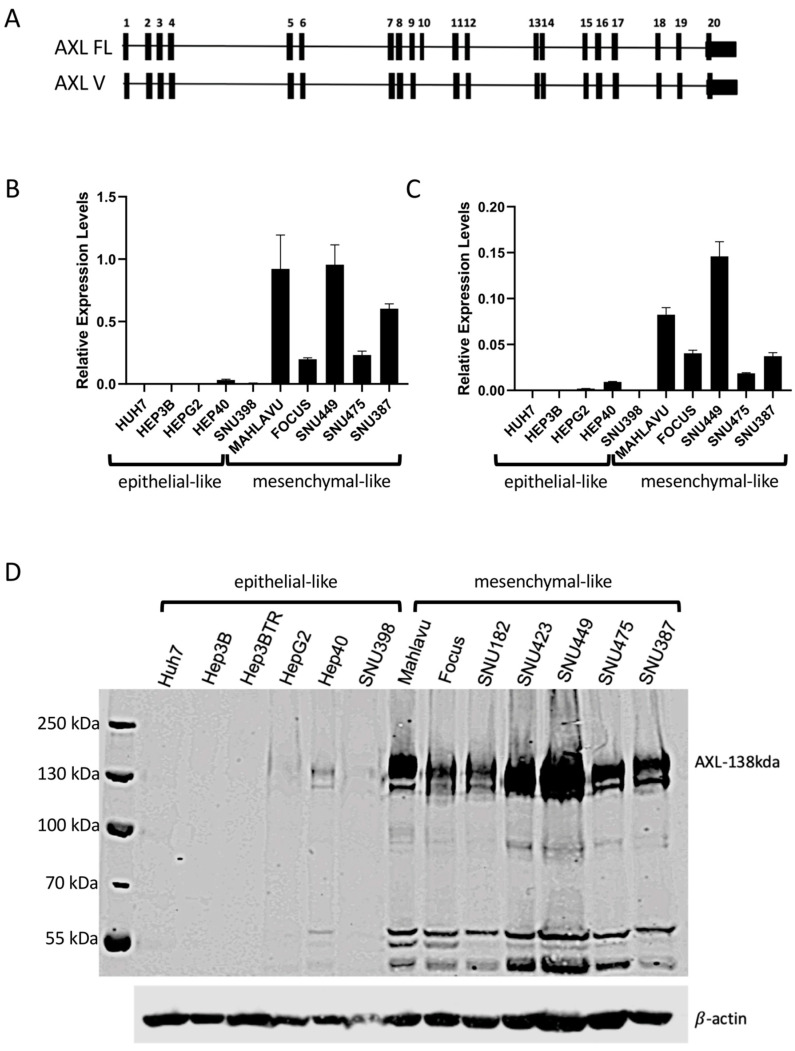
Expression Analysis of AXL in HCC Cell Lines. (**A**) The exon representations of the two alternative spliced isoforms of AXL. (**B**,**C**) mRNA levels of the long transcript, AXL FL, and short transcript, AXL V, were examined in 10 HCC cell lines using the real-time polymerase chain reaction (qPCR) technique. (**D**) Total proteins were analyzed in 13 HCC cell lines. 138 kDa AXL protein and shorter AXL protein forms were detected in mesenchymal-like but not epithelial-like HCC cell lines. β-actin was used as a loading control.

**Figure 2 ijms-22-13247-f002:**
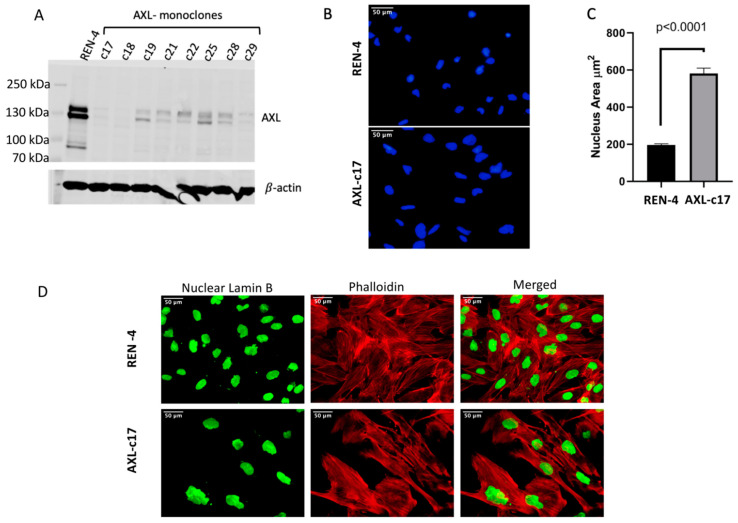
Expression levels of AXL in monoclones, REN-4 and AXL-c17, and changes in morphology and nucleus size. (**A**) 138 kDa AXL expression was detected in REN-4 negative control cells and AXL-monoclones except for the c17 and c18. β-actin served as a loading control. (**B**) Nucleus staining by DAPI (blue) in REN-4 (top) and AXL-c17 (bottom). (**C**) Nuclear areas were calculated by Image J program. Significantly larger nucleus sizes were observed in AXL-c17 (*p* < 0.0001) (**D**) Phalloidin (red) which stains F-actin protein, highlights the differences in cell morphology and nuclear lamin B (green) shows the nucleus differences.

**Figure 3 ijms-22-13247-f003:**
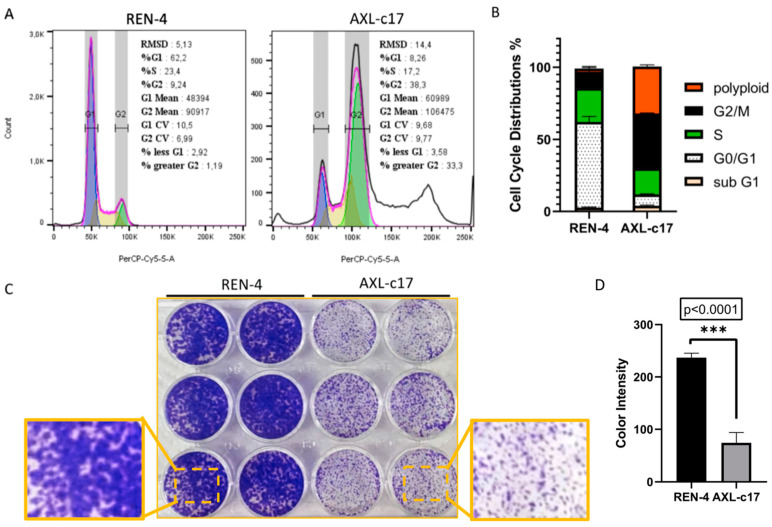
Cell cycle and colony formation of control and AXL knock-out cells. (**A**) DNA content from cell cycle analysis of REN-4 (left) and AXL-c17 (right) were shown. (**B**) Percentage of cell cycle distribution of both REN-4 and AXL-c17 cells. (**C**) Colony formation of REN-4 and AXL-c17 cells. Left 6 wells, and right 6 wells depict REN-4 and AXL-c17 cells, respectively. The focused area was displayed with dashed lines. (**D**) The color intensities of colony formations of REN-4 (black) and AXL-c17 (gray) were calculated by Image J. (*** *p* > 0.0001).

**Figure 4 ijms-22-13247-f004:**
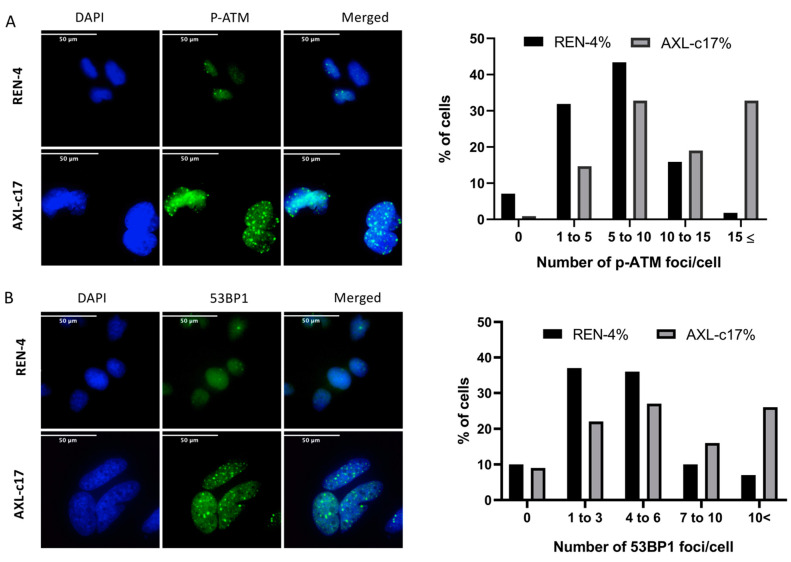
Foci formation as a marker of DNA damage response in AXL knock-out cells. Representative immunofluorescent image (left) and number of foci per cell distribution (right) of (**A**) p-ATM (Ser1981) and (**B**) 53BP1. Approximately 300 cells were counted for quantification. DAPI (blue) stains nucleus and foci were shown with green dots. The 53BP1 and p-ATM foci number per single cell was grouped as 0, 1–3, 4–6, 7–10, and greater than 10, and 1–5, 5–10, 10–15, and greater than 15, respectively.

**Figure 5 ijms-22-13247-f005:**
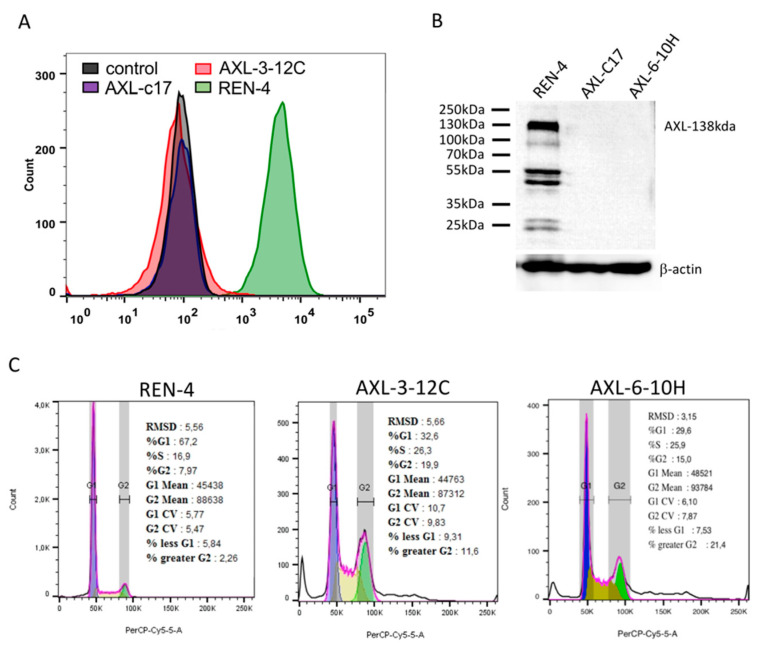
AXL expression and cell cycle analysis of newly obtained AXL knock-out clones, AXL-3-12C and AXL-6-10H. (**A**) AXL expression in REN-4, AXL-3-12C, AXL-c17 cells were analyzed by flow cytometry. Unstained control cells were used as a negative control. (**B**) AXL expressions of REN-4, AXL-c17, AXL-6-10H were analyzed by western blot. β-actin was used as a loading control. (**C**) Cell cycle distribution of control REN-4 and newly obtained AXL knock-out clones AXL-3-12C, AXL-6-10H, respectively.

**Figure 6 ijms-22-13247-f006:**
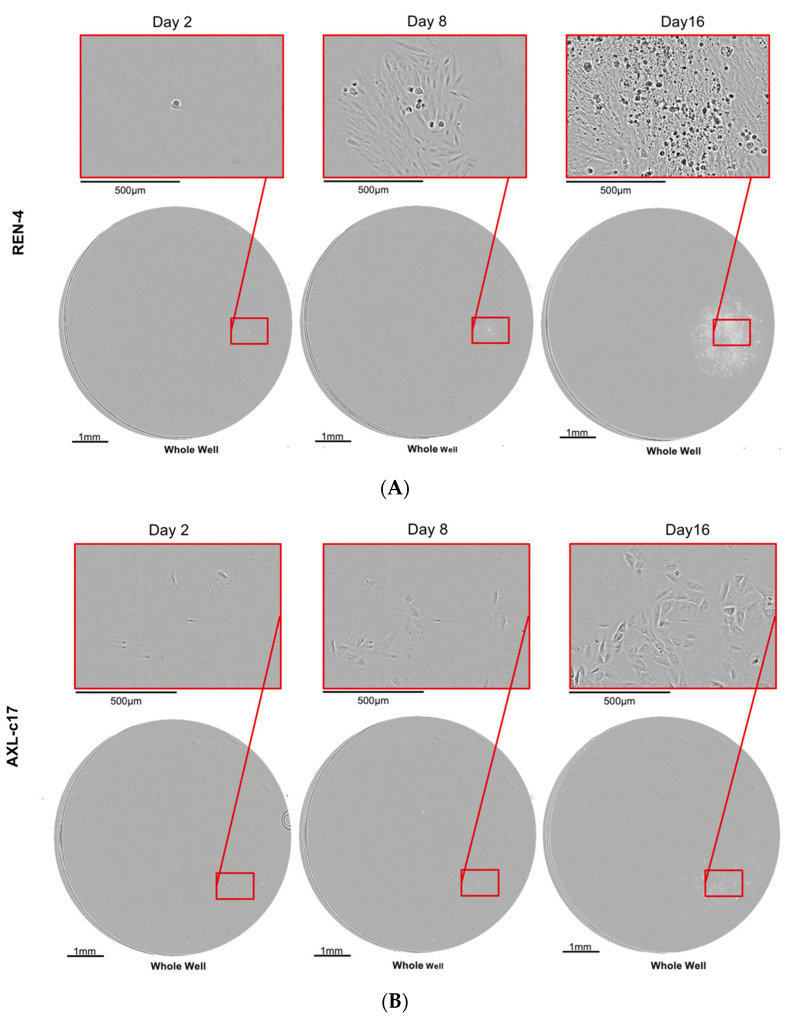
Clonality-proliferation assay and cell cycle distribution of single-cell clones REN-4, AXL-c17, AXL-6-10H. (**A**–**C**) Representative clonality-proliferation images of REN-4, AXL-c17, AXL-6-10H at days 2, 8, 16, respectively. (**D**) The cell cycle analyses of randomly selected clones from clonality-proliferation assay.

**Figure 7 ijms-22-13247-f007:**
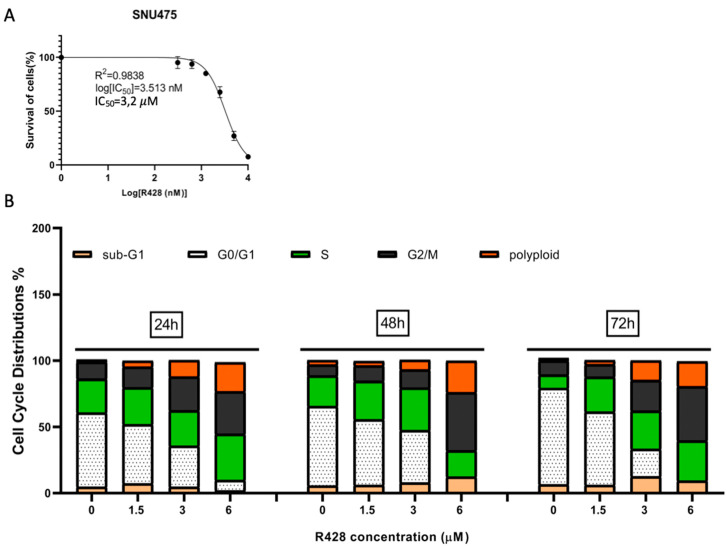
G2 cell cycle arrest was detected in AXL inhibitor R428-treated parental SNU475 cell line. (**A**) Cell survival curve versus log R428 concentration (IC50 = 3.2 µM). (**B**) Cell cycle distribution of parental SNU475 cells versus increasing concentration of R428 at 24, 48, 72 h treatment. Sub-G1, G0/G1, S, G2/M, polyploid cells were colored in light orange, white, green, black, and orange, respectively.

**Figure 8 ijms-22-13247-f008:**
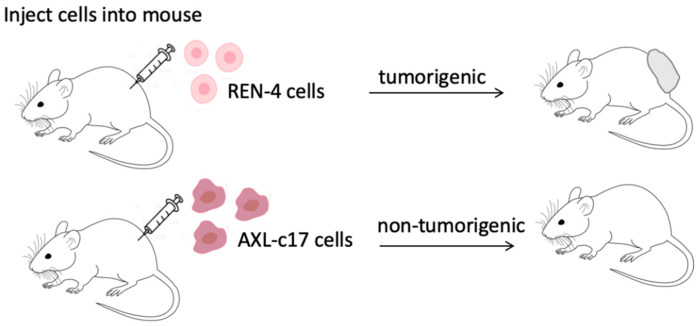
The scheme of animal experiments. Subcutaneous injection of control REN-4 cells induces tumor formation, whereas injection of AXL-c17 cells results in no tumor formation in all mice.

**Table 1 ijms-22-13247-t001:** Results of Tumor Formation experiments. The number of animals and injected cell number in each group is listed.

Injected Cell Number	Number of Animals with Tumors REN-4	Number of Animals with Tumors AXL-c17
1 × 10^6^	1/1	0/1
2 × 10^5^	4/4	0/3
Total	5/5	0/4

## Data Availability

Data supporting the result are available at Izmir Biomedicine and Genome Center archives.

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
