# Peer review of "AXL Knock-Out in SNU475 Hepatocellular Carcinoma Cells Provides Evidence for Lethal Effect Associated with G2 Arrest and Polyploidization"

_ijms, 2021, doi:10.3390/ijms222413247_

Round 1

Reviewer 1 Report

In this study the Authors analyzed the mechanism of action of AXL in HCC. Their findings  indicate that AXL expression is closely correlated with a mesenchymal phenotype in HCC cell lines. Analyzing the role of AXL in mesenchymal SNU475 cells by CRISPR-Cas9 mediated gene knock-out was showed a drastic phenotypic changes, including increased DNA damage response, prolongation of doubling time, G2 arrest, and polyploidization in vitro and loss of tumorigenicity in vivo. Pharmacological inhibition of AXL by R428 recapitulated G2 arrest and polyploidy phenotype. These observations suggest that acute loss of AXL in some mesenchymal HCC cells is lethal and point out that its inhibition may represent a druggable vulnerability in AXL-high HCC patients.

Comments and suggestions:

-The Authors should define the features of the AXL antibody. Especially, defining which variants of the AXL protein was linked by the AXL antibody. Or if it links both AXL protein variants.

-In Material and Methods section describe all cell lines used in the experiments and characterize them.

Author Response

Dear Reviewer, 

Best Regards, 

Reviewer 2 Report

Dear Authors,

In my opinion, this is well written article, with use of really nice English language (congratulations :)). The subject is extremely important and connects with one of the most important matters in HCC molecular basics and usefulness of newest immunochemotherapeutic agents ... However, as a reviewer, I have some comments & concerns:

1. the main potential problem for me is the structure of the text ... I mean the chronology & composition of chapters could be improved a bit - materials&methods chapter is at the very end of paper and earlier we have all the results -> to describe the results You needed then some additional comments on methods that had been used to achieve those results ... and some times You even compile some fragments of discussion in the results chapter ... but although - everything is substantively OK ... I will give some examples: 

lines 167-9: "... we performed series of analyses to explore DNA damage response ..." - rather for methods ?

lines 169-71: "... DNA damage response is complex process, which involves damage sensors ..." - rather for introduction & aim ?

lines 191-2: "Findings reported here with AXL-c17 clone may represent the cellular effects of AXL gene inactivation& off-targets effects ..." - rather for discussion ?

lines 193-5: "We performed additional viral infection experiments on SNU475 cells using lentiviral vectors expressing gRNA, ..." - rather for methods ?

2. lack of some abbreviations at the first use in the text - examples:

EMT - Epithelial-Mesenchymal Transition 

DAPI - DiAmidinePhenylIndole

... we have to remember, that not every potential reader is an expert in this subject :)

3. minor style&meaning errors - examples: line 25 (abstract) - "In vitro and loss of tumorigenicity in vivo." ??? / line 302 - "... the expression AXL ..." - rather: the expression of AXL (or) the AXL expression ???

Best Wishes :)

Author Response

Dear Reviewer, 

Best Regards, 

Reviewer 3 Report

In the present research, the authors show that Axl expression is upregulated in hepatocellular carcinoma and associated mesenchymal subtype. Axl knockdown was associated with overall growth-suppressive effects.  The authors also employ pharmacological inhibition of Axl for potential clinical translation. Although the authors provide solid evidence on the involvement of Axl in hepatocellular carcinoma, the results should be inferred with caution. My specific comments are appended as below:

Major concerns:

  1. In the initial screening, do authors use any non-tumorigenic control line to examine the expression of Axl?
  2. Axl KO- how do authors verify the positive CRISPR clones?
  3. Animal studies- describe the methodology in more detail. What was the age of animals? Were they randomized before treatment?
  4. Figure 2D: there seems a difference in the scale in upper and lower panels. It should be uniform.
  5. Figure 3C- authors should try to quantitate the results, either counting or dissolving the stain in 10% acetic acid and taking readouts at 590.
  6. Figure 7A: authors should show inhibition with other positive and negative control lines shown in figure 1.
  7. Animal experiment: I am not totally convinced with the data shown. It deserves a separate figure with details as a scheme of the experiment, dose used and the relevant information. In addition, do authors perform any toxicological analysis as ,measurement of animal weight/ AST levels?

Minor concerns:

  1. Introductions section- please share details on median survival on treatment.
  2. Western blot, IF- note the dilutions of antibodies used.

Author Response

Dear Reviewer, 

Best Regards, 

Round 2

Reviewer 3 Report

I congratulate the authors for providing the modifications. I suggest to take care of following minor points:

  1. Animal experiment- present the scheme of experiment in the figure.
  2. Authors should provide the HE images if have slides.
  3. In the discussion section, note a paragraph on the limitations of this study.

Author Response

Dear Reviewer,

We are glad that you are pleased with the modification that we made. Concerning the minor revision you requested, please find our responses:

  1. Animal experiment- present the scheme of the experiment in the figure.

The scheme of the animal experiment has now been added as Figure 8 in the revised manuscript

  1. Authors should provide the HE images if have slides.

 We kept paraffin blocks of tumors. However, we don’t have slides either stained or unstained. Due to time constraints, we are not able to provide H&E staining pictures which ,we believe,  have limited value due to the fact that AXL knock-out cells did not form any tumors.

  1. In the discussion section, note a paragraph on the limitations of this study.

We revised the manuscript accordingly (please find in the last paragraph of the discussion)

Best Regards